# Tunable Electronic and Optical Properties of MoGe$_2$N$_4$/AlN and MoSiGeN$_4$/AlN van der Waals Heterostructures toward Optoelectronic and Photocatalytic Applications

Jingyao Shao [1], Jian Zeng [1], Bin Xiao [1,*], Zhenwu Jin [1], Qiyun Wang [1], Zhengquan Li [1,*], Ling-Ling Wang [2], Kejun Dong [3] and Liang Xu [1,2,*]

[1] Jiangxi Provincial Key Laboratory for Simulation and Modelling of Particulate Systems, School of Energy and Mechanical Engineering, Jiangxi University of Science and Technology, Nanchang 330013, China; 15062135197@163.com (J.S.); zj905721748@163.com (J.Z.); 19979416377@163.com (Z.J.); 17379935188@163.com (Q.W.)

[2] Key Laboratory for Micro-Nano Optoelectronic Devices of Ministry of Education, School of Physics and Electronics, Hunan University, Changsha 410082, China; llwang@hnu.edu.cn

[3] Centre for Infrastructure Engineering, School of Engineering, Design and Built Environment, Western Sydney University, Penrith, NSW 2751, Australia; kejun.dong@westernsydney.edu.au

[*] Correspondence: xiaobin@jxust.edu.cn (B.X.); zhengquan.li@jxust.edu.cn (Z.L.); liangxu@hnu.edu.cn (L.X.)

**Abstract:** Van der Waals (vdW) heterostructures provide an effective strategy for exploring and expanding the potential applications of two-dimensional materials. In this study, we employ first-principles density functional theory (DFT) to investigate the geometric, electronic, and optical properties of MoGe$_2$N$_4$/AlN and MoSiGeN$_4$/AlN vdW heterostructures. The stable MoGe$_2$N$_4$/AlN heterostructure exhibits an indirect band gap semiconductor with a type-I band gap arrangement, making it suitable for optoelectronic devices. Conversely, the stable MoSiGeN$_4$/AlN heterostructure demonstrates various band gap arrangements depending on stacking modes, rendering it suitable for photocatalysis applications. Additionally, we analyze the effects of mechanical strain and vertical electric field on the electronic properties of these heterostructures. Our results indicate that both mechanical strain and vertical electric field can adjust the band gap. Notably, application of an electric field or mechanical strain leads to the transformation of the MoGe$_2$N$_4$/AlN heterostructure from a type-I to a type-II band alignment and from an indirect to a direct band transfer, while MoSiGeN$_4$/AlN can transition from a type-II to a type-I band alignment. Type-II band alignment is considered a feasible scheme for photocatalysis, photocells, and photovoltaics. The discovery of these characteristics suggests that MoGe$_2$N$_4$/AlN and MoSiGeN$_4$/AlN vdW heterostructures, despite their high lattice mismatch, hold promise as tunable optoelectronic materials with excellent performance in optoelectronic devices and photocatalysis.

**Keywords:** van der Waals heterostructures; electronic structure; photocatalytic; first-principles calculation



## 1. Introduction

Environmental pollution poses significant threats to human survival and development as it continues to escalate. The depletion of carbon-based fossil fuels, along with their increasing consumption, highlights the urgent need for renewable energy alternatives with exceptional performance. Consequently, there is ongoing exploration of novel materials and technologies to address these pressing needs [1]. Notably, since 2004, the successful isolation of graphene from three-dimensional materials has marked the beginning of a new era in two-dimensional nanomaterial research [2]. Building upon this breakthrough, new two-dimensional materials have been continuously produced and studied, including graphene, hexagonal boron nitride (h-BN), and transition metal dichalcogenides (TMDs), among others [3–7]. Moreover, van der Waals (vdW) heterostructures, formed by vertically

stacking two-dimensional layered materials, have been extensively explored in theory and experiment [8]. Based on the alignment of the conduction and valence bands of two materials, two-dimensional semiconductor van der Waals heterostructures can be categorized into three types: type-I (straddling gap) heterojunctions, where both the Valence Band Maximum (VBM) and the Conduction Band Minimum (CBM) are located within the same semiconductor material, type-II (staggered gap) heterojunctions, in which the VBM and CBM originate from different semiconductors, and type-III (broken gap) heterojunctions, where there is an energy crossover between the VBM and CBM [8–12]. Therefore, combining different single-layer two-dimensional materials in a composite structure may yield structural, electronic, and optical properties superior to those of isolated materials, thus significantly expanding the design space and functionality of two-dimensional materials. These advancements play a crucial role in catalysis, optoelectronic detection, and optoelectronic devices [13–18].

Additionally, two-dimensional AlN can be successfully prepared between graphene and Si substrates using metal organic chemical vapor deposition (MOCVD) technology, exhibiting a planar hexagonal structure similar to graphene [19–26]. Research indicates that AlN holds enormous potential for application in optical devices owing to its ability to absorb photons in the ultraviolet and visible light ranges [27]. However, its large indirect bandgap, lower carrier mobility, and poor photoelectric response performance pose obstacles to its broader utilization [28]. Fortunately, the construction of a vdW heterostructure offers the possibility of breakthrough performance. For instance, studies have shown that $AlN/MoSe_2$ and $AlN/WS_2$ heterostructures all exhibit band gaps suitable for photocatalytic water splitting [29].

Moreover, recent advancements in materials synthesis have led to the production of a novel centimeter-level single-layer $MoSi_2N_4$ thin film with excellent performance via chemical vapor deposition [27,30,31]. This development holds significant promise for future nanodevices and catalysis applications. Additionally, the exploration of the two-dimensional $MA_2X_4$ material family—where M = Mo, W, V, Nb, Ta, Ti, Zr, Hf, or Cr, A = Si or Ge, and X = N, P, or As—has sparked considerable research interest [32,33]. Notably, the prediction of a new Janus $MoSiGeN_4$ monolayer film with good stability, a suitable band edge position, and excellent light absorption ability suggests potential applications in various fields [34–39]. Nguyen et al. reported the use of graphene and $MoSiGeN_4$ to form heterostructures to prepare high-performance nanoelectronic devices [40]. Lv et al. studied the structural and electronic properties of different stacking configurations of double-layer $MoSiGeN_4$ and achieved a type-II alignment electronic structure through the built-in electric field [41]. Wang et al. reported the use of the $MoGe_2N_4/MoSTe$ heterostructure as a promising tunable optoelectronic material [42]. Inspired by the above content, we decided to combine $MoSiGeN_4/MoGe_2N_4$ with AlN to construct heterostructures and study the changes in their electronic properties through biaxial strain and a vertical electric field.

Therefore, for the purpose of this paper, $MoGe_2N_4/AlN$ and $MoSiGeN_4/AlN$ vdW heterostructures were constructed. Each heterostructure adopts six different high-symmetry stacking modes. The geometric structure and electronic properties of AlN, $MoSiGeN_4$, and $MoGe_2N_4$ monolayers were verified by density functional theory (DFT). It was found that $MoGe_2N_4/AlN$ exhibits different band gap arrangements in different stacking modes, while $MoSiGeN_4/AlN$ was identified as a type-I heterostructure, suitable for optoelectronic devices. Subsequently, we tuned the electronic properties of the heterostructure through biaxial strain and electric field. The results demonstrate that the band gap energy can be effectively adjusted under the influence of plane biaxial strain and electric field while maintaining the intrinsic type-II band alignment. This work contributes to further advancements in the research of $MA_2Z_4$ family materials and showcases their potential applications in optoelectronic and nanoelectronic devices, offering significant value for practical applications.

## 2. Computational Methodology

The first-principles calculations based on density functional theory (DFT) were performed using the Vienna ab initio simulation package (VASP) [43–45]. The Perdew–Burke–Ernzerhof (PBE) generalized gradient approximation (GGA) was used to describe the electron exchange and related information. The projection enhanced wave method of plane wave basis set was used to describe the interaction between electrons and ions [46,47]. The weak vdW force between layers was corrected using the empirical correction method of grimme (DFT-D3) [48]. We also solve the problem of band gap underestimation by comparing the more accurate electronic properties of the 2006 HSE06 hybrid functional with the PBE method [49]. The energy cut-off value was set to 500 eV. The Brillouin zone was calculated using the Monkhorst–Pack k point grid $9 \times 9 \times 1$. The structural relaxation was calculated using the joint gradient method. The convergence accuracy of ion motion was set to 0.01 eV/Å, and the self-consistent convergence accuracy of electron force calculation was set to $10^{-8}$ eV [50]. A 25 Å vacuum space was used along the z direction to solve possible periodic interactions. Visualization of all structures was attained using VESTA [51].

## 3. Results and Discussion

Firstly, the geometric and electronic structures of AlN, $MoGe_2N_4$, and $MoSiGeN_4$ monolayers were studied. The construction of AlN, $MoGe_2N_4$, and $MoSiGeN_4$ monolayers was based on previously reported experimental and calculated lattice parameters [52–54]. Figure 1a presents the top and side views of the geometric structure of the AlN monolayer, revealing its hexagonal structure. The band gap, calculated as 3.78 eV using the HSE06 method, aligns with the value of 2.93 eV obtained by the PBE method, consistent with existing literature [52]. For the $MoGe_2N_4$ monolayer depicted in Figure 1b, its geometric configuration is illustrated, with its original cell comprising 1 Mo, 2 Ge, and 4 N atoms. The $MoGe_2N_4$ monolayer is characterized as an indirect semiconductor with a band gap of 1.27 eV (PBE calculated value of 0.94 eV) [47]. Figure 1c,f display the geometric configuration and band structure of $MoSiGeN_4$, with a band gap of 1.75 eV (PBE calculated value of 1.37 eV) [55]. Additionally, Figure 1d–f present the band diagrams obtained from HSE06 functional calculations for AlN, $MoGe_2N_4$, and $MoSiGeN_4$ monolayers, showcasing their indirect band gaps of 3.78 eV, 1.27 eV, and 1.75 eV, respectively. These values are consistent with previous experimental and theoretical studies [43,56,57]. In summary, while some properties of these three monolayers exhibit potential for improvement, several stacked structures were constructed using quantum methods, with strain distributed between the $MoGe_2N_4$/AlN and $MoSiGeN_4$/AlN layers.

As shown in Table 1, the fully relaxed bond lengths of AlN and $MoGe_2N_4$ monolayers were 1.78 Å for Al-N and 2.121 Å for Mo-N, respectively. Considering $MoSiGeN_4$ as a Janus two-dimensional material, the bond lengths of Mo-N1 and Mo-N2 were 2.101 Å and 2.104 Å, respectively, consistent with previous research results [58,59]. Furthermore, the relaxed lattice constants of AlN, $MoGe_2N_4$, and $MoSiGeN_4$ were estimated at 3.120 Å, 3.021 Å, and 2.956 Å, respectively [40,43,56].

**Table 1.** The optimized lattice constant a (Å), interlayer distance d (Å), interlayer binding energy $E_b$ (eV), and band gap with HSE06 function, $E_{HSE06}$ (eV).

| Structure | a (Å) | d (Å) | $E_b$ (eV) | $E_{HSE06}$ (eV) |
|---|---|---|---|---|
| AlN | 3.120 | - | - | 3.783 |
| $MoGe_2N_4$ | 3.021 | - | - | 1.273 |
| $MoSiGeN_4$ | 2.956 | - | - | 1.753 |
| AB5 ($MoGe_2N_4$/AlN) | 3.006 | 2.852 | −0.016 | 1.382 |
| AC5 ($MoSiGeN_4$/AlN) | 3.055 | 2.637 | −0.196 | 0.886 |

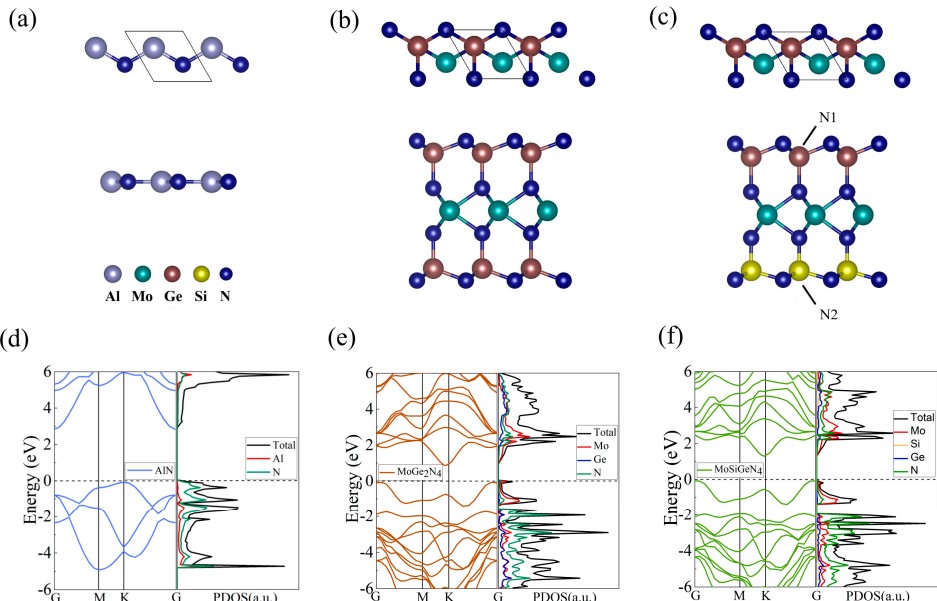

**Figure 1.** The geometric structure of (**a**) AlN, (**b**) MoGe$_2$N$_4$, and (**c**) MoSiGeN$_4$ monolayers. The calculated band structure and its DOS and PDOS of (**d**) AlN, (**e**) MoGe$_2$N$_4$, and (**f**) MoSiGeN$_4$ monolayers. The Fermi level was set to zero.

The binding energy is one of the effective methods to evaluate the stability of heterostructures. Through this approach, we calculate the binding energy (E$_b$) of heterostructures using the following formula:

$$E_b = E_{vdW} - E_{AlN} - E_{either} \qquad (1)$$

Here, E$_{vdW}$ is the energy of the MoGe$_2$N$_4$/AlN or MoSiGeN$_4$/AlN heterostructure, E$_{AlN}$ is the energy of the AlN monolayer, and E$_{either}$ is the energy of the MoGe$_2$N$_4$ or MoSiGeN$_4$ monolayer. According to the equation, the negative E$_b$ value indicates that the energy of the heterostructure is stable, with a more negative value suggesting greater stability. Therefore, the stacking method with a negative binding energy and relatively low interfacial adhesion energy was selected.

Given the similarity in lattice constants among these three monolayers, a lattice mismatch rate of less than 5% can be achieved without the establishment of a supercell. The chosen vdWs heterostructures consist of the AlN cell, MoGe$_2$N$_4$ cell, and MoSiGeN$_4$ cell. Additionally, considering the differences in structural stacking, we investigated six high-symmetry stacking methods for research purposes. For these MoGe$_2$N$_4$/AlN (Figure 2a) and MoSiGeN$_4$/AlN (Figure 2b) heterostructures, a comparative approach was employed, such as utilizing the same stacking method for AB1 and AC1. As two-dimensional materials extend infinitely on the plane, we examined the top view and side view of these six geometric structures, detailing each structure comprehensively. Under the convergence criterion, all structures underwent geometric optimization concerning energy and force. The results revealed that the MoGe$_2$N$_4$/AlN heterostructure exhibits a band gap ranging from 1.64 eV to 1.76 eV (Figure 3a). Notably, the fifth stacking mode (AB5) demonstrated a type-II heterostructure with a direct band gap of 0.89 eV. Under solar illumination, electrons in the VBM of the AlN or MoGe$_2$N$_4$ layer were excited and transitioned to their CBM, leaving behind holes in their VBM. Subsequently, driven by the built-in electric field between layers, electrons in the CBM of MoGe$_2$N$_4$ layer transferred and accumulated toward the CBM of the AlN layer. Similarly, holes accumulated in the VBM of the MoGe$_2$N$_4$ layer. Electrons and holes accumulated in different layers, effectively suppressing the recombination of photo-generated electrons and holes, enhancing its photocatalytic activity. On the other hand, the MoSiGeN$_4$/AlN heterostructure displays an indirect band gap ranging from

1.26 eV to 1.42 eV (Figure 3b), smaller than the band gaps of 1.75 eV and 3.78 eV of the $MoSiGeN_4$ and AlN monolayers, respectively. Due to the relatively small lattice mismatch rate of the heterostructure and the negative interface adhesion energy corresponding to the weak vdW bonding between the constituent layers, successful experimental preparation is feasible. The fifth stacking method was adopted for both the $MoGe_2N_4/AlN$ and the $MoSiGeN_4/AlN$ heterostructures to study the impact of applied strain and electric field on their electronic structures.

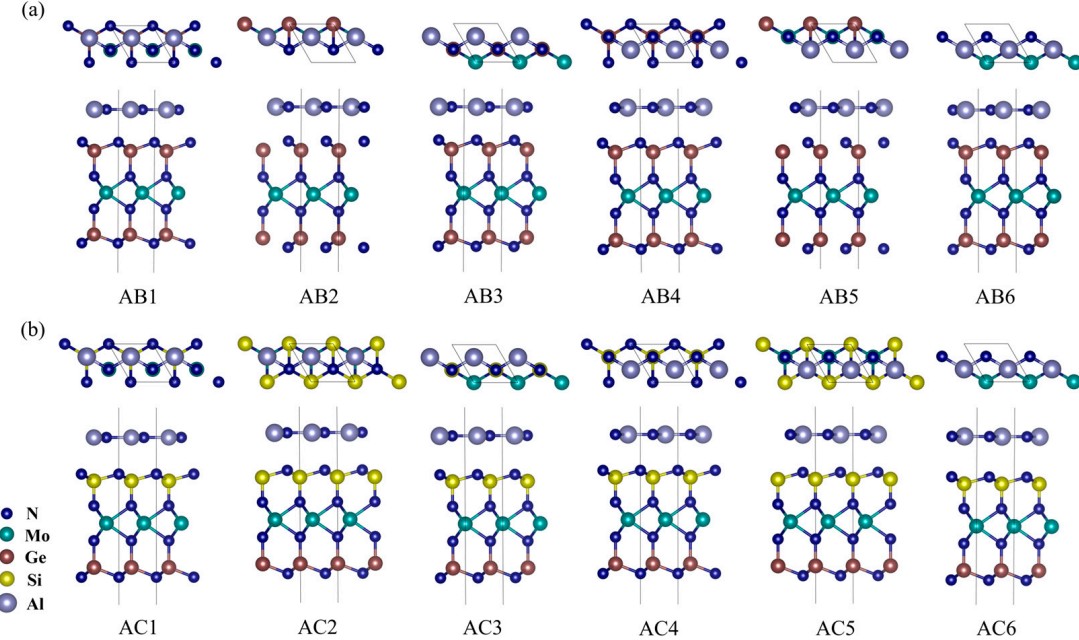

**Figure 2.** Top and side views of optimized (**a**) AB1, AB2, AB3 AB4, AB5, and AB6 different stacking modes of $MoGe_2N_4/AlN$; (**b**) AC1, AC2, AC3, AC4, AC5, and AC6 different stacking modes of $MoSiGeN_4/AlN$.

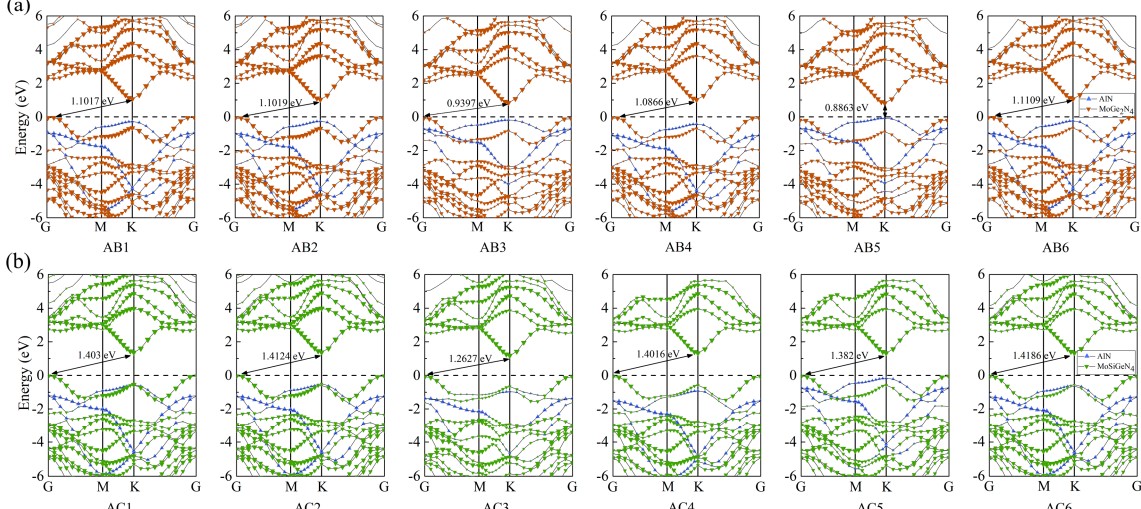

**Figure 3.** The band structure of six different stacking heterostructures relative to (**a**) $MoGe_2N_4/AlN$ and (**b**) $MoSiGeN_4/AlN$. The Fermi level was set to zero.

To delve deeper into the electronic properties of the $MoGe_2N_4/AlN$ and $MoSiGeN_4/AlN$ heterostructures, we generated three-dimensional charge density difference plots for

these two heterostructures, as illustrated in Figure 4. The calculation formula is defined as follows:

$$\Delta\rho = \rho_{vdW} - \rho_{AlN} - \rho_{either} \tag{2}$$

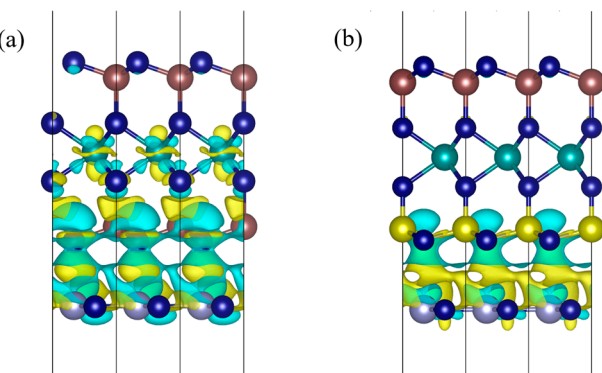

**Figure 4.** The 3D charge density difference of heterostructures (**a**) MoGe$_2$N$_4$/AlN and (**b**) MoSiGeN$_4$/AlN. The orange and green regions represent charge accumulation and consumption, respectively.

Here, $\rho_{vdW}$ and $\rho_{AlN}$ denote the total charge density of the heterostructure and the charge density of the AlN monolayer, respectively, and $\rho_{either}$ denotes the charge density of the MoGe$_2$N$_4$ or MoSiGeN$_4$ monolayer. The resulting three-dimensional charge density difference is depicted in Figure 4, where the green region signifies charge depletion and the orange region signifies charge accumulation. As depicted in Figure 4, charge transfer is evident at the interfaces of the MoGe$_2$N$_4$/AlN and MoSiGeN$_4$/AlN heterojunctions.

Furthermore, we conducted the Bader charge analysis of the heterostructure to quantify the charge transfer, as shown in Table 2. The results indicate that 0.0486 electrons were transferred from the MoGe$_2$N$_4$ layer to the AlN layer in the MoGe$_2$N$_4$/AlN heterostructure, while 0.0187 electrons were transferred from the MoSiGeN$_4$ layer to the AlN layer in the MoSiGeN$_4$/AlN heterostructure. This electron transfer led to the formation of a built-in electric field at the interface of the heterostructure.

**Table 2.** The Bader charge analysis of AB5(MoGe$_2$N$_4$/AlN) and AC5(MoSiGeN$_4$/AlN) vdW heterostructures. The gain and loss electrons are represented by negative and positive values, respectively.

| Structure | | AB5(MoGe$_2$N$_4$/AlN) | | | AC5(MoSiGeN$_4$/AlN) | | |
|---|---|---|---|---|---|---|---|
| | AlN | −0.0486 | Al | −2.3120 | AlN | −0.0187 | Al | −2.3095 |
| | | | N | +2.2634 | | | N | +2.2908 |
| charge (e) | MoGe$_2$N$_4$ | +0.0486 | Mo | −1.5126 | MoSiGeN$_4$ | +0.0187 | Mo | −1.5062 |
| | | | Ge | −1.8670 | | | Si | −2.9019 |
| | | | N | +1.3238 | | | Ge | −1.8054 |
| | | | | | | | N | +1.5580 |

Next, we proceeded by investigating the impact of biaxial strain on the electronic characteristics of the MoGe$_2$N$_4$/AlN and MoSiGeN$_4$/AlN vdW heterostructures. Biaxial strains ranging from −5% to 5%, incremented by 1%, were applied to the heterostructures. Figure 5 illustrates the band gap profiles of the heterostructures under varying biaxial strains. It is evident that the electronic properties of the MoGe$_2$N$_4$/AlN heterostructures are sensitive to biaxial strain. With an increment in biaxial strain from 0 to 5%, the band gap of the heterostructure was found to exhibit a diminishing trend, transitioning from a type-II direct band gap to a type-I indirect band gap at a strain of 4%. Conversely, as the biaxial strain decreased incrementally, the band gap value displayed an increasing trend. The heterostructure maintained a type-II direct band gap as the strain diminished,

leading to an augmentation in the band gap value. Similarly, the band gap behavior of the MoSiGeN$_4$/AlN heterostructure was found to align with that of MoGe$_2$N$_4$/AlN, demonstrating a linearly decreasing trend with increasing biaxial strain. Conversely, as the biaxial strain decreased incrementally, the band gap value exhibited a linearly increasing trend. Notably, under a −4% biaxial strain, the heterostructure transitioned from a type-I indirect band gap to a type-II direct band gap. Additionally, it can be inferred from Figure 5 that when the direct band gap was of type-II, the valence band maximum (VBM) of both the MoGe$_2$N$_4$/AlN and the MoSiGeN$_4$/AlN heterostructures was contributed to by AlN. Conversely, when it was a type-I indirect band gap, the VBM and conduction band minimum (CBM) of these two heterostructures originated from MoGe$_2$N$_4$ and MoSiGeN$_4$, respectively.

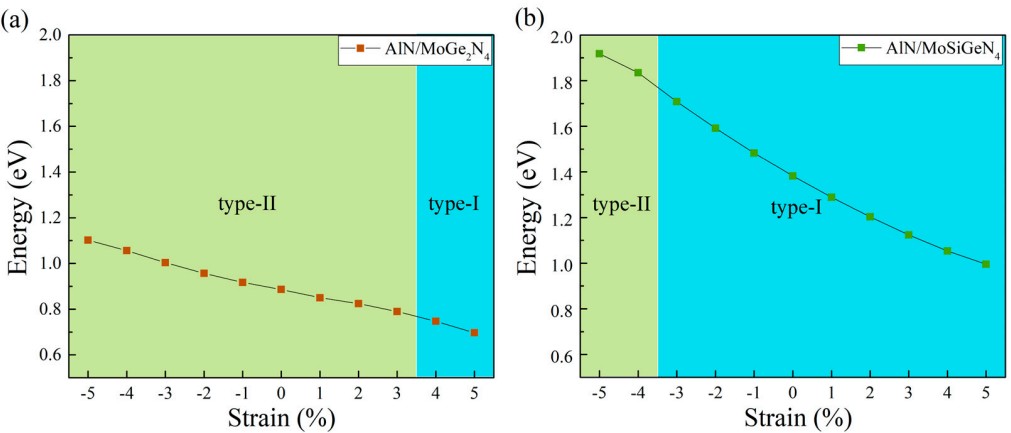

**Figure 5.** The band gap plots of (**a**) AlN/MoGe$_2$N$_4$ and (**b**) AlN/MoSiGeN$_4$ as a function of the strain.

To tailor the band gap of heterostructures to nanoelectronic devices, we explored the effect of an electric field on their electronic properties. The electric field applied to the heterostructures ranged from −0.3 V/Å to 0.3 V/Å, incremented by 0.1 V/Å. Figure 6 illustrates the band gap variation of the heterostructures under the influence of the electric field. The band gap of the MoGe$_2$N$_4$/AlN heterostructure was found to exhibit a linear increase in the range of −0.3 V/Å to 0.3 V/Å. Notably, the heterostructure transitioned from a type II direct band gap to a type-I indirect band gap at +0.1% V/Å. Conversely, the band gap value of the MoSiGeN$_4$/AlN heterostructure initially increased and then decreased with the application of the electric field. Specifically, a change in the band gap type occurred when the electric field was set to −1% V/Å. In the range of −0.3% V/Å to −0.1% V/Å, the heterostructure demonstrated a type-II direct band gap. Remarkably, the band gap types of these two heterostructures can be modulated by the applied electric field, offering the possibility of adjusting both the MoGe$_2$N$_4$/AlN and the MoSiGeN$_4$/AlN heterostructures. The band gap characteristics of the heterojunctions provide valuable theoretical insights for experimentalists to effectively engineer photocatalytic hydrogen production heterojunctions of two-dimensional materials and integrate them into optoelectronic devices.

Expanding beyond band gap considerations, the optical absorption spectra of AlN, MoGe$_2$N$_4$, and MoSiGeN$_4$ monolayers, along with MoGe$_2$N$_4$/AlN and MoSiGeN$_4$/AlN heterostructures, were analyzed as important indicators for photovoltaic device materials, as depicted in Figure 7. Different colors represent various two-dimensional materials. Dashed lines illustrate the optical absorption spectra of individual monolayers, while solid lines depict the optical absorption spectra of the constructed heterostructures. Notably, both the MoGe$_2$N$_4$/AlN and the MoSiGeN$_4$/AlN heterostructures were found to exhibit significantly stronger absorption of the visible light compared to the single-layer AlN, MoGe$_2$N$_4$, and MoSiGeN$_4$, effectively compensating for the deficiencies in visible light absorption of the AlN monolayer. Moreover, the peak absorption coefficients of these two heterostructures were both close to $30 \times 10^5$ cm$^{-1}$. This enhancement also renders

these heterostructures superior with respect to optical properties compared to certain other photovoltaic materials [60]. This heightened absorption capability suggests that these heterostructures possess enhanced potential for efficiently utilizing solar energy, rendering them promising candidates for photovoltaic applications. The observed optical characteristics provide valuable insights for experimentalists, facilitating the effective engineering of photocatalytic hydrogen production heterojunctions using two-dimensional materials and their integration into advanced optoelectronic devices.

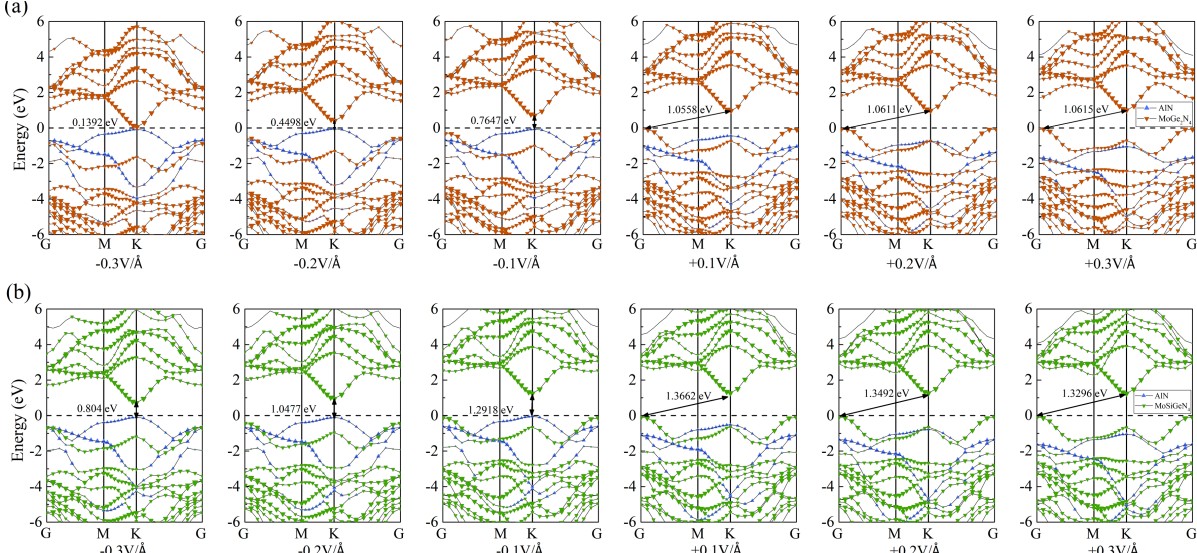

**Figure 6.** The band structure of (**a**) MoGe$_2$N$_4$/AlN and (**b**) MoSiGeN$_4$/AlN under the action of an external electric field, with corresponding external electric field values at the bottom. The Fermi level was set to zero.

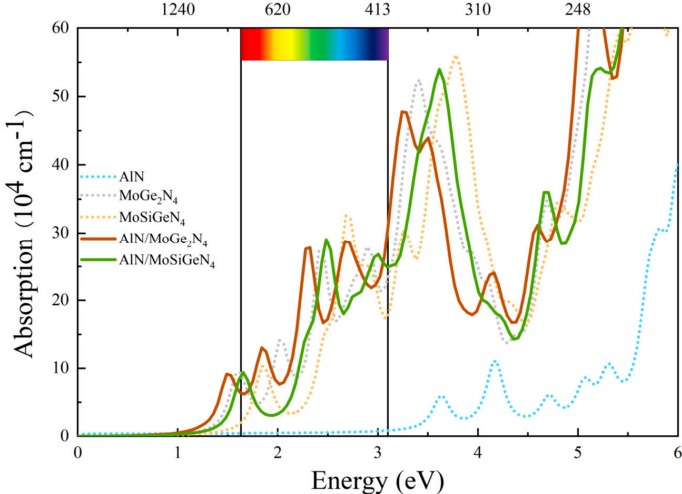

**Figure 7.** The optical absorption spectra of isolated monolayers and heterostructure. The ribbon area represents the visible light absorption range.

## 4. Conclusions

In summary, we investigated the electronic properties of the MoGe$_2$N$_4$/AlN and MoSiGeN$_4$/AlN van der Waals heterostructures through density functional theory simulations, examining their responses to biaxial strain and vertical electric fields. Our findings reveal that different stacking methods significantly impact the electronic properties of these heterostructures. While all six high-symmetry structures of the MoSiGeN$_4$/AlN heterostructures exhibit type-I indirect band gaps, the situation differs for the MoGe$_2$N$_4$/AlN

heterostructures, among which the AB5 configuration manifests a type-II direct band gap. Notably, for a given heterostructure, the band gap exhibits fluctuations within a certain range. Under the influence of biaxial strain and electric fields, the band gap type of the heterostructure undergoes effective transformation. Particularly, the band gap value of the $MoSiGeN_4/AlN$ heterostructure exhibits relatively significant changes under biaxial strain. In the presence of an electric field ranging from $-0.3\,V/Å$ to $-0.1\,V/Å$, the $MoSiGeN_4/AlN$ heterostructures transition from a type-I indirect band gap to a type-II direct band gap. These findings underscore the potential for effectively adjusting the electronic properties of the $MoGe_2N_4/AlN$ and $MoSiGeN_4/AlN$ heterostructures through strain and applied electric fields, enabling modifications of the band gap type. Such insights offer valuable theoretical guidance for the efficient preparation and utilization of these heterostructures, indicating promising applications in optoelectronic devices and photocatalysis.

**Author Contributions:** Conceptualization, B.X., Z.L. and L.X.; Methodology, J.S.; Validation, J.S., B.X., Z.J. and Q.W.; Formal analysis, Z.L., K.D. and L.X.; Investigation, J.Z.; Data curation, J.S.; Writing—original draft, J.S.; Writing—review & editing, J.Z., Q.W., K.D. and L.X.; Visualization, J.S., B.X. and L.X.; Supervision, B.X., Z.L., L.-L.W., K.D. and L.X.; Project administration, L.X.; Funding acquisition, L.X. All authors have read and agreed to the published version of the manuscript.

**Funding:** This work was financially supported by the National Nature Science Foundation of China (Grant No. 52263031), the Jiangxi Provincial Natural Science Foundation (Grant Nos. 20212BAB201013, 20202ACBL211004), and the Open Project Program of Jiangxi Provincial Key Laboratory for Simulation and Modelling of Particulate Systems, Jiangxi University of Science and Technology.

**Institutional Review Board Statement:** Not applicable.

**Informed Consent Statement:** Not applicable.

**Data Availability Statement:** Data are contained within the article.

**Conflicts of Interest:** The authors declare no conflict of interest.

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
