# Peer review of "Tunable Electronic and Optical Properties of MoGe2N4/AlN and MoSiGeN4/AlN van der Waals Heterostructures toward Optoelectronic and Photocatalytic Applications"

_coatings, doi:10.3390/coatings14040500_

Round 1

Reviewer 1 Report

Comments and Suggestions for Authors

The authors employ first-principles density functional theory (DFT) to investigate the electronic and optical properties of MoGe2N4/AlN and MoSiGeN4/AlN vdW heterostructures. They explore the effect of stacking order, mechanical strain and vertical electric field on the electronic properties of these heterostructures.

The paper is interesting for researchers working in the field of 2D materials, but it requires a few revisions before publication

·        As the authors state “AlN is prepared between graphene and Si substrates using metal organic chemical vapor deposition (MOCVD) technology”. Is it possible to isolate the AlN monolayer and realize experimentally the MoGe2N4/AlN and MoSiGeN4/AlN vdW heterostructures? If not the effect of substrates must be taken into account in the theoretical modelling.

·        What exactly the authors mean for “total energy of the heterostructure” (line 137) where the formula (1) comes from?

·        In figure 2 the top view of the stacking modes it is not clear, i.e., the Al sites are not visible.

·       Is it possible to experimentally control the proposed stacking modes? Have the authors considered to change the relative angle?

·       How does it change the absorption spectrum of the two heterostructures with stacking order and strain?

Comments on the Quality of English Language

minor editing

Reviewer 2 Report

Comments and Suggestions for Authors

Employing first-principles density functional theory (DFT), Shao et al. investigate the geometric, electronic, and optical properties of MoGe2N4/AlN and MoSiGeN4/AlN van der Waals (vdW) heterostructures. The stable MoGe2N4/AlN heterostructure exhibits an indirect band gap with a type-I arrangement, suitable for optoelectronic devices. Conversely, the stable MoSiGeN4/AlN heterostructure displays various band gap arrangements depending on stacking modes, making it suitable for photocatalysis applications. Mechanical strain and vertical electric fields are found to adjust the band gap, leading to transformations in band alignments.

Comments:

1. Authors should discuss type-I and type-II vdW heterostructures with references in the introduction.

2. “…the optical absorption spectra of AlN, 240 MoGe2N4, and MoSiGeN4 monolayers, along with MoGe2N4/AlN and MoSiGeN4/AlN het-241 heterostructures, are analyzed as important indicators for photovoltaic device materials, as 242 depicted in Fig. 7.” Authors should explain the optical properties in details in the manuscript and compare it with other photovoltaic materials.

3. Authors should elucidate the rationale behind the suitability of type-II MoGe2N4/AlN heterostructures for photocatalytic applications.
